# Are Roma People Descended from the Punjab Region of Pakistan: A Y-Chromosomal Perspective

**DOI:** 10.3390/genes13030532

**Published:** 2022-03-17

**Authors:** Atif Adnan, Allah Rakha, Hayder Lazim, Shahid Nazir, Wedad Saeed Al-Qahtani, Maha Abdullah Alwaili, Sibte Hadi, Chuan-Chao Wang

**Affiliations:** 1Department of Anthropology and Ethnology, Institute of Anthropology, School of Sociology and Anthropology, Xiamen University, Xiamen 361000, China; 2Department of Forensic Sciences, College of Criminal Justice, Naif Arab University of Security Sciences, Riyadh 11452, Saudi Arabia; walqahtani@nuass.edu.sa (W.S.A.-Q.); shadi@nauss.edu.sa (S.H.); 3Department of Forensic Sciences, University of Health Sciences, Lahore 54600, Pakistan; dnaexpert@me.com (A.R.); s.nazir@uhs.edu.pk (S.N.); 4Faculty of Health, Social Care and Medicine, Edge Hill University, Ormskirk L39 4QP, UK; alazawihayder@yahoo.com; 5Department of Biology, College of Sciences, Princess Nourah Bint Abdulrahman University, Riyadh 84428, Saudi Arabia; maalwaele@pnu.edu.sa

**Keywords:** Gypsies, Y-chromosomal STRs, Pakistan, migration, Europe

## Abstract

Gypsies are a separate ethnic group living in Pakistan and some other countries as well. They are mostly known as ‘Roma’ and ‘untouchables’. They have different types of lifestyles as compared to other common people, as they always keep migrating from one place to another. They do not have proper houses; they live in tent houses and most probably work on daily wages to earn their living. Gypsies cannot be specified according to the place of residence and can only be classified according to their migration route. Previous historical and linguistic research showed the north Indian origin of Roma people. The present study collected 285 unrelated Roma individuals living in Punjab and typed with the Goldeneye Y20 system. Allelic frequencies ranged between 0.0035 and 0.5266, with haplotype diversity (HD) of 0.9999 and discrimination capacity (DC) of 0.8790. Gene diversity (GD) ranged from 0.6489 (DYS391) to 0.9764 (DYS391) (DY385ab). A total of 223 unique alleles were observed. Interestingly, the haplogroup R accounted for 40.56% and J for 22.06%. In MDS analysis, Pakistani Roma formed a close cluster with Roma from Constanta, Romania. The migration pattern of the Roma population from Pakistan, India and Europe was inferred using coalescence theory in the Migrate-n program. Overlapping Y-STR data were used to test different migration models. These migration models showed us the dominant gene flow from Pakistan to India and Europe to Pakistan. The results of our study showed that Y STRs provided substantially stronger discriminatory power in the Pakistani Roma population.

## 1. Introduction

In general, it is believed Gypsies or Roma people began their journey from northern India (modern-day Pakistan) through several migrations, and they were in Persia (modern-day Iran and Turkey) by the 11th century [1]. At the beginning of the 14th century, they were in southeastern Europe and by the 15th century, they were in western Europe. By the 18th century, they had travelled to America, and today, they live all over the world. Their population size is approximately 10–15 million, while they are the largest ethnic group in eastern Europe. Some Roma people still live in the traditional manner, migrating from place to place, always staying outside of the cities, while others have joined the larger society around them. In every place they have ever lived, the Roma people have taken up local languages and religions, married into the local population and somehow retained their distinct identity [2]. The Roma language belongs to the Indo-European language family, has more than 60 dialects and does not have a single convention for writing. Because of their diverse nature, they do not have a written history; therefore, experts can only infer their history through linguistics, historical records of other nations they contacted with and through genetic investigations [3]. Anthropological records showed conspicuous resemblances between the cultures of the different Indian groups and Roma people [4]. Roma people have the same social structure, which is practiced in India, where the groups are usually defined by profession [5,6]. In previous centuries, Roma people practiced endogamy like many other Indian populations. One clan of Roma people would get married within their clan (sub-ethnic group) and across clans; marriages were not entertained or welcomed, much as Indian Brahmins would get married to Brahmins only, or a Shudra could not get married to a Brahmin [7]. Previous studies found similarities between the Roma and Banjara (Indian wandering Gypsy tribe) residents of southern and central India [8]. The migration route of Roma people is still unknown, so lingual influence was used as a supplementary tool to better understand their migration route. According to Grierson et al. [3], Romani is of ‘Dardic origin’, but according to Turner et al. [9], it belongs to the Indo-Aryan subgroup as Urdu/Punjabi/Hindi. Counting in Romani from ‘one to ten’ is the same as in Urdu, along with several other words, such as ‘ankh’, ‘naak’, kaan’, ‘paani’. The presence of Burushaski [10], Dardic, Georgian, Ossetian, Armenian and mediaeval Greek language words and shortage or absence of Arabic words in Roma language [2] also supports the northerly route of migration, starting around the Gilgit Baltistan along the southern Caspian coast, the southern border of the Caucasus and the Black Sea, through the Bosporus and later on across Europe [11]. There is another folklore about the migration of Gypsies, namely that they migrated to Europe about a thousand years ago from the areas of Sindh, Punjab and Rajasthan. In the 5th century, the ruler of Persian Sassanid monarchy, Bahram Gul, imported around 10,000 musicians and dancers from Hindustan (Sindh, Gujrat and Punjab regions) and established their colonies across Persia. According to scientific theories [12], many of these ‘ancient Meerasis’ continued their migration to Byzantine and Europe and are the predecessors of Gypsies. The largest tribes of Gypsies settled in central Europe and call themselves ‘Roma’. Their cousins in northern Europe call themselves ‘Zintis’, which could be derived from ‘Sindhis’, since their ancestors migrated from around the Sindhu River [12]. The features of the Roma people stand out in Europe. They look more like Indians and Pakistanis than other populations, though they are usually more rugged and defiant.

Genetic studies based on uniparental markers, such as mitochondrial DNA and Y-chromosome, established the south Asian origin of the Roma population. The prevalence of Y-chromosomal haplogroup H1a [13,14] and mtDNA haplogroups M18, M35b and M5a1 in Roma populations [15,16] gives us a hint of their south Asian ancestry. Nevertheless, studies based on the Y-chromosome and mtDNA contradict each other. Y-chromosomal studies suggest that Gypsies originate from south India, while mtDNA studies suggest their northern Indian ancestry [16]. Previous studies covering fourteen European Roma groups covering genome-wide data suggest that the Romani people originated in north/northwestern India (modern-day Pakistan) 1500 years ago [17,18]. A recent study found that the Roma ancestors diverged from the Punjabi population of northwest India [19].

In this study, we sampled 285 unrelated Roma individuals across Punjab, Pakistan, intending to investigate the possible origin of the Pakistani Roma population by utilizing the Y-chromosome short tandem repeats (STRs) to (i) shed light on the genetic makeup of the Pakistani Roma population, (ii) their relationship to its close-neighboring population and (iii) whether the European Roma people are descended from the Punjab region of Pakistan.

## 2. Materials and Methods

### 2.1. Samples

A total of 285 blood samples were collected from different areas of Punjab, Pakistan. All participants who were included in this study were unrelated and self-declared Roma/Gypsy individuals of at least three generations. All participants gave their informed consent either orally and with a thumbprint (in case they could not write) or in writing after the study aims and procedures were carefully explained to them. This study was approved by the University of Health Sciences Lahore Pakistan (2019-CMU-1/04). All the experimental procedures were performed by the standards of the Declaration of Helsinki 1964.

### 2.2. Laboratory Procedures

Axygen AxyPrep Blood Genomic DNA Miniprep Kit was used to extract genomic DNA according to the manufacturer’s protocol (Axygen Biosciences; Union City, USA), and the final concentration of DNA was diluted to 1 to 2 ng/μL. DNA was amplified using GoldenEye Y20 system Kit (PeopleEye, Beijing, China). PCR amplification was carried out using the Applied Biosystems GeneAmp PCR System 9700 thermal cyclers. PCR amplifications were performed as recommended by the manufacturer, although using half of the recommended reaction volume (12.5 μL). After successful PCR amplification, the PCR products were analyzed by using an 8 capillary ABI 3500 DNA Genetic Analyzer with POP-4 polymer (Life Technologies, CarIsbad, CA, USA) according to the manufacturer’s protocol. GeneMapper I-DXSoftware version 1.4 (Life Technologies, CarIsbad, CA, USA) was used for the genotype assignment. DNA typing was performed according to the manufacturer’s protocol using the locus panel and allele bins supplied by the manufacturer and allele designations corresponding with the allele ladder supplied by the manufacturer. Genotype nomenclature was based on the International Society for Forensic Genetics recommendations. Our laboratory participated and passed the YHRD quality assurance exercise in 2015. Haplotype data were already made accessible via the Y-chromosome Haplotype Reference Database (YHRD) under accession number YA004554.

### 2.3. Statistical Analysis

Allelic and haplotype frequencies were computed by direct counting method, and haplotype diversity (HD) was calculated according to
$$\mathrm{HD}=\left(1-\sum_{i}p_{i}^{2}\right)$$
where *n* is the male population size, and *p_i_* is the frequency of *i*-th haplotype. Discrimination capacity (DC) was calculated as the ratio of unique haplotypes in the samples. Match probabilities (MP) were calculated as
$$\left(\sum_{i}p_{i}^{2}\right)$$
where *p_i_* is the frequency of the *i*-th haplotype. We also evaluated these forensic genetics parameters at 9 Y STR loci of minimal haplotype (DYS19, DYS389I, DYS389II, DYS390, DYS391, DYS392, DYS393, and DYS385ab), 11 loci of extended haplotype (DYS19, DYS389I, DYS389II, DYS390, DYS391, DYS392, DYS393, DYS385ab, DYS438 and DYS439), Powerplex 12 Y (11 extended haplotype loci + DYS437), Yfiler™ (12 PPY loci + DYS448, DYS456, DYS458, DYS635, Y_GATA_H4) and finally at 20 Y STRs (17 Yfiler loci + DYS388, DYS447 and DYS460). Genetic distances were evaluated using the Rst, Fst statistic and to perform analysis of molecular variance (AMOVA) tests between reference populations and currently studied populations on overlapping STRs, (DYS19, DYS389I, DYS389II, DYS390, DYS391, DYS392, DYS393, DYS437, DYS438, DYS439, DYS448, DYS456, DYS458, DYS635, and Y_GATA_H4) were calculated by using Arlequin Software v3.556 [19]. We calculated both Rst and Fst values because, in the generalized stepwise mutation model, Rst offers relatively unbiased evaluation of migration rates and times of population divergence, while on other hand, Fst tends to show too much population similarity, predominantly when migration rates are low or divergence times are long [20]. Reduced dimensionality spatial representation of the populations was performed based on Rst values using multi-dimensional scaling (MDS) with IBM SPSS Statistics for Windows, Version 23.0 (IBM Corp., Armonk, NY, USA).

A neighbor-joining phylogenetic tree was constructed for the Roma and the reference populations based on a distance matrix of Fst using the Mega7 software [21]. We also predicted Y haplogroups in the Roma population using the Y-DNA Haplogroup Predictor NEVGEN (http://www.nevgen.org (accessed on 10 January 2022)). Any haplotypes that had null alleles or duplication variants in the Roma population were excluded from the analyses.

#### 2.3.1. The Median-Joining Network

To define the genetic structure of the Roma population, a median-joining network was constructed. We used the stepwise mutation model and median joining-maximum parsimony algorithm [22] by using the program Network-5 as described at the Fluxus Engineering website (http://www.fluxus-engineering.com (accessed on 10 January 2022). The weighting criteria for Y-STRs were set following Ref. [23]. Any haplotypes that had null alleles or duplication variants in the Roma population were excluded from the analysis.

#### 2.3.2. Assessment of Gene Flow and Migration in Roma Population

We used MIGRATE program version 4.2.14 [24] by applying coalescence theory to determine the migration rates between the Roma population from Pakistan and other relevant populations from India, Turkey and Europe. Population genetics parameters were calculated using the Bayesian inference. With the sampling increment, a wider search of genealogy can be attained. So, we ran a long chain with a sampling increment of 1000. The burn-in value was set at 5000 for discarded trees per chain. According to the number of discarded trees and increment value, each sample was reviewed 5000 times (P. Beerli, personal communication). The metropolis-coupled MCMC or ‘heating’ algorithm was applied for supplementary searches with appropriate acceptance criteria [25,26,27]. We used the heating menu to ensure that at least four chains were available. A shortcut to specifying the temperatures is available on the menu. It generates temperatures that are spaced in a specific way. They are spaced so that the temperature’s inverse is regularly spaced on the 0 to 1 interval. For example, the temperatures in the four chains we chose are 1.0, 1.5, 3.0, 100,000.0, resulting in spacing of 1.0 (1/1.0), 0.666 (1/1.5), 0.333 (1/3) and 0.0 (1/100,000). This was implemented to improve the program’s search approach and aid in the discovery of the optimal tree (P. Beerli, personal communication). The hotter chains move more freely and explore more genealogy space than the cold chains. Input data files were prepared using the PGD Spider data converting tool [28]. The rate of gene flow was explored at two levels: level one was within the subcontinent, while level two was movement from Pakistan to Europe via Turkey. The first model gives us information about the direct migration from one population to the other; the second one informed about the separation from the ancestral population; the third one informed about the further divergence with ongoing migration from the ancestral population; while the fourth one would assume that two populations belonged to the same panmictic population and was only used in level three. To generate the Bayes factor, log marginal likelihood with different runs was used and, subsequently, Bayes factors were used for model comparisons so that their scales would inform about the population differences.

## 3. Results and Discussion

### 3.1. Genetic Diversity

A total of 281 out of 285 individuals were successfully genotyped, and their genotypes are summarized in Appendix A. A total of 247 haplotypes were observed at 20 Y STRs, and among these 247 haplotypes, 223 (79.35%) were unique haplotypes or singletons. The haplotype diversity (HD) was 0.9999, discriminatory capacity (DC) was 0.8790, and random matching probability (RMP) was 0.0036. When we reduced the numbers of STRs from 20 to 9, several haplotypes also reduced to 207, with 176 (68.75%) singletons. The HD was 0.99774, DC was 0.8085, and RMP was 0.0065. The locus or haplotype diversity (GD) and the number of observed alleles or haplotypes for 20 STRs are summarized in Appendix A. Among single-copy Y STRs, DYS391 showed the lowest gene diversity (0.6489) with 6 different allele combinations, while DYS390 showed the highest gene diversity (0.8562) with 13 different allele combinations among single-copy Y STRs. Goldeneye 20Y contained one multi-copy STR (DYS385), which not only showed the highest diversity (0.9764) but also the highest number of allele combinations (69) among 19 Y STRs. Allele frequency in the Roma population ranged from 0.0071 to 0.5266.

### 3.2. Population Pairwise Rst and Fst Distance Comparisons

The genetic distances (Rst and Fst) between the Roma population and other relevant populations from Pakistan, India, Afghanistan and Europe are listed in Appendix A. The relationship between the Roma population and European Roma population is poorly described, so we used two different methods based on their similarity with a priori expectations. Fst is a standardized variance of haplotype frequency and assumes genetic drift as being the agent that differentiates populations. Rst is a standardized variance of haplotype size and takes into account both drift and mutation as causes of population differentiation, assuming a stepwise model in which each mutation creates a new allele, either by adding or deleting a single repeat unit. According to Rst pairwise genetic distances, the Romanian population from Constanta, Romania, (Rst = −0.0106) showed the closest distance to Roma Pakistani population, which was followed by another Romanian population (Rst = 0.0049) from Transylvania, Romania. In subcontinental population Afridi Pathan (Rst = 0.0189) population from Uttar Pradesh, India, and followed by Indian population (Rst = 0.0275) from Rajasthan, India. According to Fst pairwise genetic distances, Indian population (Fst = 0.0006) from Rajasthan, India, showed the closest genetic distance, which was followed by the Romanian population (Fst = 0.0008) from Germany. The results of these genetic distances are in support of that theory, which says that the Roma population dispersed from Punjab and Sindh [12].

### 3.3. MDS and Neighbor-Joining Tree Analyses

Phylogenetic relationships among the Roma Pakistani population and 80 other regional reference populations were assessed using MDS analysis based on Rst distances derived from the Y-STR data (Appendix A). In this MDS plot, the bulk of populations were placed on the central left side, while Roma populations were placed in the lower bottom and formed a loose cluster. We also assessed the phylogenetic relationship only among Roma populations from Pakistan and Europe using the MDS plot. Pakistani Roma population placed itself on the upper right corner of the plot, along with German, Roma and Italian Roma populations (Appendix A and Figure 1). In the neighbor-joining trees, an admixed population will always lie on the path between the source populations [29]. We observed sixteen clusters from down to up, in which Roma and non-Roma groups were separated. Such a separation is statistically significant, and most of the Roma populations were in the third cluster, while the Pakistani Roma population was in the ninth cluster (Appendix A). In first, second, third and fourth branches, we found Pathan and Pashtun populations from Afghanistan, Pakistan and India. In fifth, sixth and seventh cluster, we found Punjabi, Kashmiri and European Roma populations. These results also give us a hint about the origin of the Roma population in the Punjab region [18,30].

### 3.4. AMOVA Analyses

AMOVA was performed on the Pakistani Roma population and other geographically and historically relevant reference populations. The results of AMOVA showed that the Hungarian Roma population has a more genetic affinity with the Pakistani Roma population. On other hand, Croatian Roma showed major genetic indifferences with the Roma population from Pakistan (Table 1).

### 3.5. Y-Chromosome Haplogroups

Ancestry informative markers (AIMs) play an important role in genealogy. So, we used NEVGEN software to calculate haplogroups from STR haplotypes. Only six haplogroups (E, G, I, J, L and R) accounted for 89% of the total Roma population from Pakistan. The median-joining network of haplotypes (Figure 2) showed the bulk of R and J haplogroups. We also presented a stacked histogram with the haplogroup composition of these populations in Appendix A.

The frequency of haplogroup E was 5.69% in the currently studied populations, and it was the most frequent haplogroup in west Asia and east Africa [31,32]. This haplogroup originated around 65 thousand years ago (KYA) [33]. This haplogroup is not found in Lithuanian or Spanish populations but is present in 4.4% of Bulgarian [34] and in 29.8% of Macedonian Roma populations [35,36]. Haplogroup G was at 8.54% in the currently studied population. It is generally believed that haplogroup G originated in eastern Anatolia, Armenia or western Iran around 48.5 KYA.

The frequency of haplogroup I was at 3.91% in the currently studied population, and this haplogroup is predominately found in central European populations. It possibly originated in central Europe around 31–35 KYA [37]. The frequency of this haplogroup was reported to be 10.3% in Portuguese [38], 25 % in Bulgarian [31], 15% in Spanish, 5% in Lithuanian and 5% in Macedonian Roma populations [35,36].

Haplogroup J accounted for 20% in the currently studied population. This haplogroup is predominately found in the Arabian Peninsula. The origin of this haplogroup is from the Middle East area known as the Fertile Crescent, comprising Palestine, Jordan, Syria, Lebanon and Iraq around 42.9 KYA [39]. Merchants from the Arabian Peninsula brought this genetic marker to the subcontinental region [40]. The frequency of this haplogroup was reported to be 33% in Portuguese [35], 21% in Spanish, 9% in Bulgarian [34] and 33% in Lithuanian Gypsies [35,38].

Haplogroup L accounts for 5% of the currently studied population, and this haplogroup is believed to have originated in the middle east or the subcontinent around 25–30 KYA [41]. The spread of this haplogroup was distributed mainly because of trade between the Arabian Peninsula and the subcontinent. This is the dominant haplogroup in Pakistani populations.

Haplogroup R originated in the north of Asia about 27 KYA (ISOGG, 2017). It is the most frequent haplogroup in Europe and Russia and, in some parts, it comprises 80% of the population. Some researchers believe that one of its branches originated in the Kurgan culture [42]. The frequency of this haplogroup is 29% in Portuguese [38], 22.2% in Spanish, 4.4% in Bulgarian [33] and 10% in Lithuanian Gypsies [35,38].

### 3.6. Estimation of Gene Flow and Migration Rate in Pakistani Roma Population

We studied Roma population gene flow at two levels. At level one, it was between the subcontinent (Pakistan and India) and at level two, from Pakistan to Europe. Publicly available data from Indian Roma, European Roma and Turkish populations were used to explore the migration route. Indian populations contained a set of west Indians, south Indians, southcentral Indians, northwest Indians, north Indians, northcentral Indians and east Indians, while the European Roma population set contained Bulgarians, Hungarians, Romanians and Slovakians. Table 2 shows level one (India→Pakistan, Pakistan→India) routes. Analysis based on Y-chromosomal migration pattern showed that model 2 (the divergence model) was the best suited model for the migration route (India→Pakistan, Pakistan→India). From India to Pakistan, the log marginal likelihood was (−1836.47, Bayes factor (−7.80), and the probability was 0.0004, while from Pakistan to India, the log marginal likelihood was (−1828.67, Bayes factor (0), and the probability was 0.9996 (Table 2). The results of level one were in concordance with Mendizabal et al. [16] and Bianco et al. [17] where they speculated that the Roma population originated in north/northwestern India (modern-day Punjab, Pakistan) around 1.5 thousand years ago (kya). At level two, the most likely route was Pakistan→Iran→Turkey→Europe, but Iranian population data were not publicly available, so we skipped Iran, and the route was Pakistan→Turkey→Europe. The most probable migration route was from Pakistan to Europe through Turkey (model 2), which yielded the log marginal likelihood of (−2533.65), Bayes factor (−4.25) and a probability of 0.0141. The surprising route was a reversal of the previous route, Europe→Turkey→Pakistan, with the log marginal likelihood of (−2529.4), Bayes factor (0) and a probability of 0.9859 (Table 3). The outcome showed the gene flow from Pakistan to India and from Europe to Pakistan. Different model approaches, which were used in this study, are shown in Appendix A. In both routes of gene flow from Pakistan to India or India to Pakistan and Pakistan to Europe or Europe to Pakistan, model 2 remained the dominant model.

## 4. Conclusions

We investigated the Y-chromosome diversity of Pakistani Roma to gain a better understanding of the Roma population structure in Pakistan and their genetic affinities with relevant local and regional populations. The dominance of the R haplogroup in the Roma population from Pakistan showed the west Eurasian ancestry like other Pakistani populations, in addition to their shared linguistic, cultural affinities, which may have impacted the paternal genetic landscape of Roma people across the world. The MDS and AMOVA results also showed the similarities between Pakistani Roma and European Roma populations. We have also explored the forensic population genetic parameters and found that the Goldeneye 20Y kit showed high discrimination power for Roma population of Pakistan, and this kit can be used for population studies and kinship testing. More comprehensive studies covering the whole genome are required to unfold information within the Pakistani Gypsy gene pool, enabling an extensive reconstruction of the major episodes in their demographic history.

## Figures and Tables

**Figure 1 genes-13-00532-f001:**
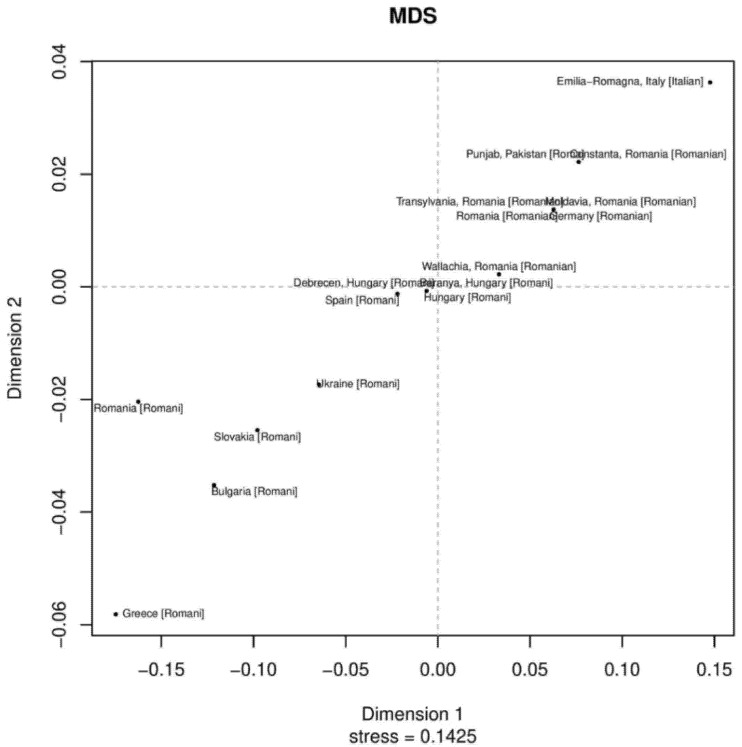
The two-dimensional plot from multi-dimensional scaling analysis of Rst values based on Yfiler haplotypes for 18 Roma populations.

**Figure 2 genes-13-00532-f002:**
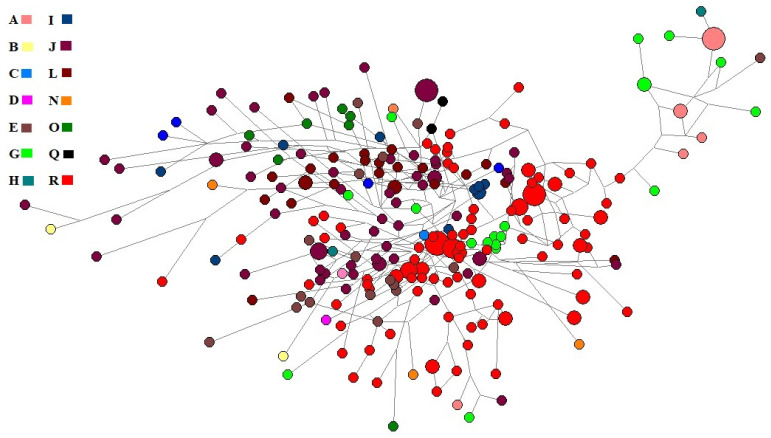
The median-joining network of the Roma population of Pakistan.

**Table 1 genes-13-00532-t001:** Analysis of molecular variance (AMOVA) using Y STRs between groups of populations.

Groups	Fst
Pakistani Roma vs. Croatian Roma	0.35647
Pakistani Roma vs. Serbian Roma	0.32266
Pakistani Roma vs. Portuguese Roma	0.29913
Pakistani Roma vs. Turkish population	0.09929
Pakistani Roma vs. Bulgarian Roma	0.09198
Pakistani Roma vs. Slovenian Roma	0.07805
Pakistani Roma vs. Romanian Roma	0.06276
Pakistani Roma vs. Hungarian Roma	0.0462

**Table 2 genes-13-00532-t002:** Level one population movements, India + Pakistan, India→Pakistan, Pakistan→India. The order of the models in each route was according to log marginal likelihood and the Bayes factor, the lowest to the highest. Log(mL), log marginal likelihood, LBF Bayes factor.

Migration Route	Model	Log(mL)	LBF	Model Probability
India + Pakistan	4	−33,811.1	−31,982.5	0
India→Pakistan	1	−28,444.7	−26,616	0
Pakistan→India	1	−26,025.5	−24,196.9	0
India→Pakistan	3	−2016.59	−187.92	0
Pakistan→India	3	−1863.45	−34.78	0
India→Pakistan	2	−1836.47	−7.8	0.0004
Pakistan→India	2	−1828.67	0	0.9996

**Table 3 genes-13-00532-t003:** Level two population movements Pakistan→Turkey→eastern Europe, eastern Europe→Turkey→Pakistan. The order of the models in each route was according to log marginal likelihood and the Bayes factor, the lowest to the highest. Log(mL), log marginal likelihood, LBF Bayes factor.

Route	Model	Log(mL)	LBF	Model Probability
Pakistan→Turkey→eastern Europe	1	−2900.02	−370.62	0
Eastern Europe→Turkey→Pakistan	1	−2851.56	−322.16	0
Pakistan→Turkey→eastern Europe	3	−2823.05	−293.65	0
Eastern Europe→Turkey→Pakistan	3	−2758.07	−228.67	0
Pakistan→Turkey→eastern Europe	2	−2533.65	−4.25	0.0141
Eastern Europe→Turkey→Pakistan	2	−2529.4	0	0.9859

## Data Availability

All necessary raw data is available in Appendix A.

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
