# Peer review of "Are Roma People Descended from the Punjab Region of Pakistan: A Y-Chromosomal Perspective"

_genes, 2022, doi:10.3390/genes13030532_

Round 1

Reviewer 1 Report

The manuscript describes population genetics analyses of Roma people from Pakistan using Y-chromosomal markers and their comparisons with available data sets. The paternal lineages found in Pakistan were described, and migration patterns from the Indian subcontinent to Europe was inferred. It contains data about the current structure of paternal lineages of Roma people and hypotheses of their historical spread, which may be interesting and valuable for researchers in the field.

However, I have some major concerns about its present form. First, I would recommend a thorough revision of the text, as there are some confusing or even incomplete sentences, that make the interpretation hard. Additionally, the list of cited references is rather short, it would be advised to include some recent articles dealing with the genetic structure and migration patterns of Roma people, for example:

Martínez-Cruz, B.; Mendizabal, I.; Harmant, C.; de Pablo, R.; Ioana, M.; Angelicheva, D.; Kouvatsi, A.; Makukh, H.; Netea, M.G.; Pamjav, H.; et al. Origins, Admixture and Founder Lineages in European Roma. European Journal of Human Genetics 2016, 24, 937–943, doi:10.1038/ejhg.2015.201.

Font-Porterias, N.; Arauna, L.R.; Poveda, A.; Bianco, E.; Rebato, E.; Prata, M.J.; Calafell, F.; Comas, D. European Roma Groups Show Complex West Eurasian Admixture Footprints and a Common South Asian Genetic Origin. PLOS Genetics 2019, 15, e1008417, doi:10.1371/journal.pgen.1008417.

Bianco, E.; Laval, G.; Font-Porterias, N.; García-Fernández, C.; Dobon, B.; Sabido-Vera, R.; Sukarova Stefanovska, E.; Kučinskas, V.; Makukh, H.; Pamjav, H.; et al. Recent Common Origin, Reduced Population Size, and Marked Admixture Have Shaped European Roma Genomes. Molecular Biology and Evolution 2020, 37, 3175–3187, doi:10.1093/molbev/msaa156.

Other suggestions regarding references can be found in the specific comments. Lastly, I find some results were poorly discussed, they would deserve a more detailed discussion. See specific comments.

Specific comments

Lines 19-21: I would suggest “Previous historical and linguistic research showed the North Indian origin of Roma people.”

Lines 24-25: I would rephrase this sentence

Lines 25-26: Consider “...formed a close cluster with Romanians from Constanta, Romania.”

Lines 37-38: "At the beginning of the 14th century, they have been reported from southeastern Europe and in the 15th century from Western Europe." Additionally, it would be beneficial to include some references about the historical migrations and distribution of Roma.

Lines 45-46: Did you mean the people or the language? It should be clearly described. I would suggest using consistent and distinguishable naming throughout the text, for example, Roma for the people and Romani for the language, or ‘Roma people’ and ‘Roma language’.

Lines 52-54: This might be true for previous centuries, but nowadays marriage is not so strictly limited, at least in Central and Eastern Europe. Additionally, as you mentioned in line 44 marriages with local people were practised long before. Thus, the sentence should be slightly modified or explained more, possibly with additional references.

Lines 56-57: Rephrase like “Previous studies found...as summarised in [8]".

Lines 57-58: I kind of understand what you meant, but linguistics is not concerned with migration per se. This sentence could be refined.

Line 60: “Romani” From this point on minor stylistic comments are omitted, as these may be superfluous after revision.

Line 70: What do you mean by “popular theories”? Scientific theories or popular anecdotes? You should refine this statement and underline it with reference.

Lines 78-80: Consider using “Prevalence of Y-chromosomal haplogroup H1a [13] and mtDNA haplogroups M18, M35b and M5a1 in Roma populations [14]...”, and adding some more references (see my general comments).

Line 87: The number of samples is 281, isn’t it?

Line 91: The number of samples is 281, isn’t it? Or is the number of blood samples collected larger than the number of samples analysed?

Lines 98-117: I find sections containing only one or two sentences unnecessary. Sections 2.2, 2.3 and 2.4 could be merged into one section named something like “Laboratory procedures”.

Lines 107-108: Please be consistent when mentioning manufacturers.

Line 144: This should be not a new subheading, rather part of “2.5.1. Phylogenetic analysis”. Additionally, when using subheadings you should follow the journals recommendations.

Line 151: This could be a new subsection: “2.5.2. Assessment of gene flow and migration in Roma populations”

Line 161: Why were these values chosen? You should justify this.

Line 174: As you did not discuss the result separately, this heading should read “3. Results and Discussion”

Lines 175, 194, 217, 236, 242 and 285: When using subheadings you should follow the journals recommendations.

Line 181: Why were the markers reduced from 20 to 9? Previously you mention 15 markers that were used for the reference populations in previous studies. It should be clearly stated which STRs were used for which analysis.

Line 220: This should refer to Figure 1, shouldn’t it?

Line 229: Numbering the branches of the phylogenetic tree this way is not informative, especially when you mention it only once. It would suffice only mention that the Pakistani Roma people clustered to a separate branch than other Roma populations. But I would like to read a bit more about the phylogenetic tree, possibly with a more detailed explanation of the branches.

Figure 1 and Figure 2: Using consistent naming for the Roma ethnic group applies also to the figures. Additionally, larger font sizes in Figure 2 could improve the readability, and maybe a larger, more detailed version of the tree could be added as a supplementary figure.

Line 248: I failed to find the supplementary figures.

Lines 251-284: I would merge this result in one subsection, probably with the title “Y-chromosome haplogroups”. Additionally, a bit more explanation of the distribution and historical patterns of the haplogroups would be helpful for the readers.

Line 292: This should refer to Table 2, shouldn’t it?

Lines 298-310: It would be helpful not only to mention ‘level one’, ‘two’ and ‘three’ when presenting the results but also to explain what these levels mean.

Line 300: the abbreviation “KYA” was used before, it should be resolved at first mention.

Lines 315-317: Please rephrase this sentence.

Lines 334-337: I failed to find the supplementary figures.

Author Response

Reviewer 1

Comments and Suggestions for Authors

The manuscript describes population genetics analyses of Roma people from Pakistan using Y-chromosomal markers and their comparisons with available data sets. The paternal lineages found in Pakistan were described, and migration patterns from the Indian subcontinent to Europe was inferred. It contains data about the current structure of paternal lineages of Roma people and hypotheses of their historical spread, which may be interesting and valuable for researchers in the field.

However, I have some major concerns about its present form. First, I would recommend a thorough revision of the text, as there are some confusing or even incomplete sentences that make the interpretation hard.

Reply: Manuscript is revised by a native speaker.

Additionally, the list of cited references is rather short, it would be advised to include some recent articles dealing with the genetic structure and migration patterns of Roma people, for example:

Martínez-Cruz, B.; Mendizabal, I.; Harmant, C.; de Pablo, R.; Ioana, M.; Angelicheva, D.; Kouvatsi, A.; Makukh, H.; Netea, M.G.; Pamjav, H.; et al. Origins, Admixture and Founder Lineages in European Roma. European Journal of Human Genetics 201624, 937–943, doi:10.1038/ejhg.2015.201.

Font-Porterias, N.; Arauna, L.R.; Poveda, A.; Bianco, E.; Rebato, E.; Prata, M.J.; Calafell, F.; Comas, D. European Roma Groups Show Complex West Eurasian Admixture Footprints and a Common South Asian Genetic Origin. PLOS Genetics 201915, e1008417, doi:10.1371/journal.pgen.1008417.

Bianco, E.; Laval, G.; Font-Porterias, N.; García-Fernández, C.; Dobon, B.; Sabido-Vera, R.; Sukarova Stefanovska, E.; Kučinskas, V.; Makukh, H.; Pamjav, H.; et al. Recent Common Origin, Reduced Population Size, and Marked Admixture Have Shaped European Roma Genomes. Molecular Biology and Evolution 202037, 3175–3187, doi:10.1093/molbev/msaa156.

Reply: All above mentioned references are added in the manuscript which gives strength to our results

Other suggestions regarding references can be found in the specific comments. Lastly, I find some results were poorly discussed, they would deserve a more detailed discussion. See specific comments. 

Specific comments

Lines 19-21: I would suggest “Previous historical and linguistic research showed the North Indian origin of Roma people.”

Reply: Revised accordingly

Lines 24-25: I would rephrase this sentence

Reply: Revised accordingly

Lines 25-26: Consider “...formed a close cluster with Romanians from Constanta, Romania.”

Reply: Revised accordingly

Lines 37-38: "At the beginning of the 14th century, they have been reported from southeastern Europe and in the 15th century from Western Europe." Additionally, it would be beneficial to include some references about the historical migrations and distribution of Roma.

Reply: Revised accordingly

Lines 45-46: Did you mean the people or the language? It should be clearly described. I would suggest using consistent and distinguishable naming throughout the text, for example, Roma for the people and Romani for the language, or ‘Roma people’ and ‘Roma language’.

Reply: revised accordingly

Lines 52-54: This might be true for previous centuries, but nowadays marriage is not so strictly limited, at least in Central and Eastern Europe. Additionally, as you mentioned in line 44 marriages with local people were practised long before. Thus, the sentence should be slightly modified or explained more, possibly with additional references.

Reply: revised accordingly

Lines 56-57: Rephrase like “Previous studies found...as summarised in [8]".

Reply: revised accordingly

Lines 57-58: I kind of understand what you meant, but linguistics is not concerned with migration per se. This sentence could be refined.

Reply: revised accordingly

Line 60: “Romani” From this point on minor stylistic comments are omitted, as these may be superfluous after revision.

Reply: revised accordingly

Line 70: What do you mean by “popular theories”? Scientific theories or popular anecdotes? You should refine this statement and underline it with reference.

Reply: revised accordingly

Lines 78-80: Consider using “Prevalence of Y-chromosomal haplogroup H1a [13] and mtDNA haplogroups M18, M35b and M5a1 in Roma populations [14]...”, and adding some more references (see my general comments).

Reply: revised accordingly

Line 87: The number of samples is 281, isn’t it?

Reply: Successfully genotyped individuals are 281, but actual collected samples were 285

Line 91: The number of samples is 281, isn’t it? Or is the number of blood samples collected larger than the number of samples analysed?

Reply: Number of blood samples collected were 285, among these 281 were genotyped successfully.

Lines 98-117: I find sections containing only one or two sentences unnecessary. Sections 2.2, 2.3 and 2.4 could be merged into one section named something like “Laboratory procedures”.

Reply: revised accordingly

Lines 107-108: Please be consistent when mentioning manufacturers.

Reply: revised accordingly

Line 144: This should be not a new subheading, rather part of “2.5.1. Phylogenetic analysis”. Additionally, when using subheadings you should follow the journals recommendations.

Reply: revised accordingly ( we have submitted the dox file which was converted to genes journal format by assistant editor and they have changed accordingly)

Line 151: This could be a new subsection: “2.5.2. Assessment of gene flow and migration in Roma populations”

Reply: revised accordingly

Line 161: Why were these values chosen? You should justify this.

Reply: revised accordingly

Line 174: As you did not discuss the result separately, this heading should read “3. Results and Discussion”

Reply: revised accordingly

Lines 175, 194, 217, 236, 242 and 285: When using subheadings you should follow the journals recommendations.

Reply: revised accordingly

Line 181: Why were the markers reduced from 20 to 9? Previously you mention 15 markers that were used for the reference populations in previous studies. It should be clearly stated which STRs were used for which analysis.

Reply: revised accordingly

Line 220: This should refer to Figure 1, shouldn’t it?

Reply: In these lines, we have discussed Roma Pakistani and 80 regional populations, and MDS plot is in supplementary figure 1

Line 229: Numbering the branches of the phylogenetic tree this way is not informative, especially when you mention it only once. It would suffice only mention that the Pakistani Roma people clustered to a separate branch than other Roma populations. But I would like to read a bit more about the phylogenetic tree, possibly with a more detailed explanation of the branches.

Reply: revised accordingly

Figure 1 and Figure 2: Using consistent naming for the Roma ethnic group applies also to the figures. Additionally, larger font sizes in Figure 2 could improve the readability, and maybe a larger, more detailed version of the tree could be added as a supplementary figure.

Reply: Figure 1 is only about Roma population (globally) based on Rst values, while figure two is about regional populations along with Roma populations which is based on Fst values.

Line 248: I failed to find the supplementary figures.

Reply: revised accordingly

Lines 251-284: I would merge this result in one subsection, probably with the title “Y-chromosome haplogroups”. Additionally, a bit more explanation of the distribution and historical patterns of the haplogroups would be helpful for the readers.

Reply: revised accordingly

Line 292: This should refer to Table 2, shouldn’t it?

Reply: revised accordingly

Lines 298-310: It would be helpful not only to mention ‘level one’, ‘two’ and ‘three’ when presenting the results but also to explain what these levels mean.

Reply: We have explained these levels in M&M section

Line 300: the abbreviation “KYA” was used before, it should be resolved at first mention.

Reply: revised accordingly

Lines 315-317: Please rephrase this sentence.

Reply: revised accordingly

Lines 334-337: I failed to find the supplementary figures.

Reply: revised accordingly

Reviewer 2 Report

  1. How did the author confirm the persons of study belong to Roma? Moreover, how other information was collected accurately along with required history. 
  2. Why the last paragraph did not define with clarity the aims of the study? 
  3. The primer sequences should be mentioned. 
  4. The phylogenetic should be based on maximum likelihood approach instead of neighbors joining approach. The use of the complex method like RAxML will be good. 
  5. The conclusion can be improve little bit by giving relevance of Pakistani with other country information. 

Reviewer 2 Report

Comments and Suggestions for Authors

  1. How did the author confirm the persons of study belong to Roma? Moreover, how other information was collected accurately along with required history. 

Reply: They were self-declared Roma population, and in consent form we have asked them about their last three generation history and according to them they are spending this life style from generation and keep moving from one place to another. Consent forms were submitted to YHRD along with genotype data.  

  1. Why the last paragraph did not define with clarity the aims of the study? 

Reply: We have written with clarity in the last paragraph of introduction 

  1. The primer sequences should be mentioned.

Reply: We used kit (GoldenEye Y20 )  

  1. The phylogenetic should be based on maximum likelihood approach instead of neighbors joining approach. The use of the complex method like RAxML will be good.

Reply: Revised accordingly  

  1. The conclusion can be improve little bit by giving relevance of Pakistani with other country information.

Reply: Revised accordingly  

Round 2

Reviewer 1 Report

The authors implemented the majority of the recommendations and included some additional references, the manuscript has been greatly improved as compared to the previous version. However, one of my questions remained mainly unanswered. This question concerned line 181 in the previous manuscript which became line 207-208 in the revised manuscript: “When we have reduced the numbers of STRs from 20 to 9”. Why were the markers reduced from 20 to 9? It should be clearly stated, ideally in the methods section, which STRs were used for which analysis and the exclusion of markers from analyses should be justified.

I find Tables 1, 2 and 3 are ill-placed. The result contained in the tables were mentioned only later in the text, tables should be closer to this part of the text. Additionally, I still find that a traditional rectangular version of the tree depicted in Figure 2 could be added as a supplementary figure.

Minor comments

Lines 61-62: I guess “The migration route of Roma people is still unknown,...” was meant.

Line 95: Consider “In this study, we have sampled 285 unrelated…”.

Lines 331-332: “Results of level one were in concordance with Mendizabal et al. [16] and Bianco et al., [41]...”

Author Response

Reviewer 1

Comments and Suggestions for Authors

The authors implemented the majority of the recommendations and included some additional references; the manuscript has been greatly improved as compared to the previous version. However, one of my questions remained mainly unanswered. This question concerned line 181 in the previous manuscript which became line 207-208 in the revised manuscript: “When we have reduced the numbers of STRs from 20 to 9”. Why were the markers reduced from 20 to 9? It should be clearly stated, ideally in the methods section, which STRs were used for which analysis and the exclusion of markers from analyses should be justified.

Reply: We have mentioned in M&M section that “We also evaluated these forensic genetics parameters at 9 Y STR loci of minimal haplotype (DYS19, DYS389I, DYS389II, DYS390, DYS391, DYS392, DYS393, and DYS385ab), 11 loci of extended haplotype (DYS19, DYS389I, DYS389II, DYS390, DYS391, DYS392, DYS393, DYS385ab, DYS438 and DYS439), Powerplex 12 Y (11 extended haplotype loci + DYS437), Yfiler™ (12 PPY loci + DYS448, DYS456, DYS458, DYS635, Y_GATA_H4) and  finally at 20 Y STRs ( 17 Yfiler loci + DYS388, DYS447 and DYS460).”

I find Tables 1, 2 and 3 are ill-placed. The results contained in the tables were mentioned only later in the text; tables should be closer to this part of the text. Additionally, I still find that a traditional rectangular version of the tree depicted in Figure 2 could be added as a supplementary figure.

Reply: I didn’t place these tables, I just submitted a dox file which was converted to Genes format by Editorial Assistant staff. Hopefully in final version they will place these tables at suitable place or when (if it gets accepted for publication) the final proofs will come to me for proof reading then I will place them near discussion of these results. Figure 2 is moved to ESM.

Minor comments

Lines 61-62: I guess “The migration route of Roma people is still unknown,...” was meant.

Reply: Revised accordingly

Line 95: Consider “In this study, we have sampled 285 unrelated…”.

Revised accordingly

Lines 331-332: “Results of level one were in concordance with Mendizabal et al. [16] and Bianco et al., [41]...”

Revised accordingly